# DSGCR: Decomposed Spectral Geometry-Aware Cross-Modal Semantic Representation for 3D Visual Grounding

**Jing He** [1]  **Licheng Jiao** [1]  **Lingling Li** [1]  **Xiaoqiang Lu** [1]  **Xu Liu** [1]  **Wenping Ma** [1]  **Fang Liu** [1]  **Long Sun** [1]

## Abstract

3D visual grounding requires robust cross-modal representation to achieve fine-grained semantic alignment and precise geometric reasoning. However, most methods employ unimodal pre-trained encoders that transfer visual and linguistic knowledge independently, inducing domain shift and poor cross-modal alignment. Meanwhile, spatial modeling with handcrafted priors limits cross-modal geometric representation, struggling to capture complex object relations due to spectral bias. To address these challenges, we propose Text-Aware Feature Tuning (TFT) and Decomposed Spectral Geometry (DSG) to enhance cross-modal semantic representation. Specifically, TFT injects linguistic context into the visual hierarchy to mitigate domain shift and facilitate early cross-modal alignment. DSG employs a learnable Fourier basis and explicitly decomposes pairwise relations into symmetric and antisymmetric spectral components, allowing the model to capture high-frequency geometric details and direction-aware relations for precise spatial reasoning. Extensive experiments on ScanRefer, Nr3D and Sr3D validate the effectiveness of our method, demonstrating state-of-the-art performance with improvements of 2.05% Acc@0.25 for 3DREC and 1.09% mIoU for 3DRES on ScanRefer.

## 1. Introduction

3D Visual Grounding (3DVG) (Chen et al., 2022b; Lin et al., 2023; Qian et al., 2024b; Chen et al., 2020; Achlioptas et al., 2020; Guo et al., 2025; Huang et al., 2021; Zhao et al., 2021; Cai et al., 2022; Wu et al., 2023; Jia et al., 2024; Li et al.,

2025) has been widely adopted in embodied AI and robotics. This task encompasses 3D Referring Expression Comprehension (3DREC) (Chen et al., 2022b; Achlioptas et al., 2020; Wu et al., 2023; Guo et al., 2025; Zhu et al., 2023; Qian et al., 2024b; Chen et al., 2020; Jia et al., 2024) and 3D Referring Expression Segmentation (3DRES) (Huang et al., 2021; Chen et al., 2025; Wu et al., 2024), aiming to locate and segment objects in 3D scenes based on natural language descriptions. Compared to 2D grounding (Xiao et al., 2024; 2023), 3D grounding must process sparse, unordered and irregular point clouds, which makes cross-modal representation learning more challenging.

Visual encoding in 3DVG typically relies on unimodal pre-trained models, achieved by either fully fine-tuning (Huang et al., 2024; 2023; Wu et al., 2023) or using frozen self-supervised models (Zhu et al., 2023). However, both methods transfer visual or linguistic knowledge independently, struggling to produce semantically aligned cross-modal representation. As shown in Figure 1(a), visual features derived from generic 3D visual-language priors (Hegde et al., 2023) are still insufficiently sensitive to semantics. Lacking guidance from textual features, the encoder has difficulty distinguishing fine-grained targets like a red seat, leading to domain shift and cross-modal misalignment. Although Parameter-Efficient Fine-Tuning (PEFT) (Jia et al., 2022; Hu et al., 2022) offers a potential solution and demonstrates strong performance in 2DVG (Xiao et al., 2024), existing 3D PEFT methods primarily target unimodal tasks such as classification (Zhou et al., 2024; Liang et al., 2025; Zha et al., 2023) and cannot be directly applied to multimodal 3DVG. Thus, we introduce text-aware tuning, which adapts PEFT to enable deep cross-modal alignment and establish robust cross-modal representation for 3D understanding.

Precise geometric reasoning is as crucial as semantic alignment for cross-modal representation. To capture spatial relations, most methods (Zhao et al., 2021; Cai et al., 2022; Chen et al., 2022b; Zhu et al., 2023; Jia et al., 2024) employ handcrafted pairwise priors such as Euclidean distances and angles. However, deep networks exhibit an intrinsic spectral bias (Rahaman et al., 2019; Xiong et al., 2025; Xu et al., 2025), tending to smooth out high-frequency geometric details while prioritizing low-frequency distance. As shown in

[1]Key Laboratory of Intelligent Perception and Image Understanding of Ministry of Education, School of Artificial Intelligence, Xidian University, Xi'an, China. Correspondence to: Licheng Jiao <lchjiao@mail.xidian.edu.cn>.

*Proceedings of the 43rd International Conference on Machine Learning*, Seoul, South Korea. PMLR 306, 2026. Copyright 2026 by the author(s).

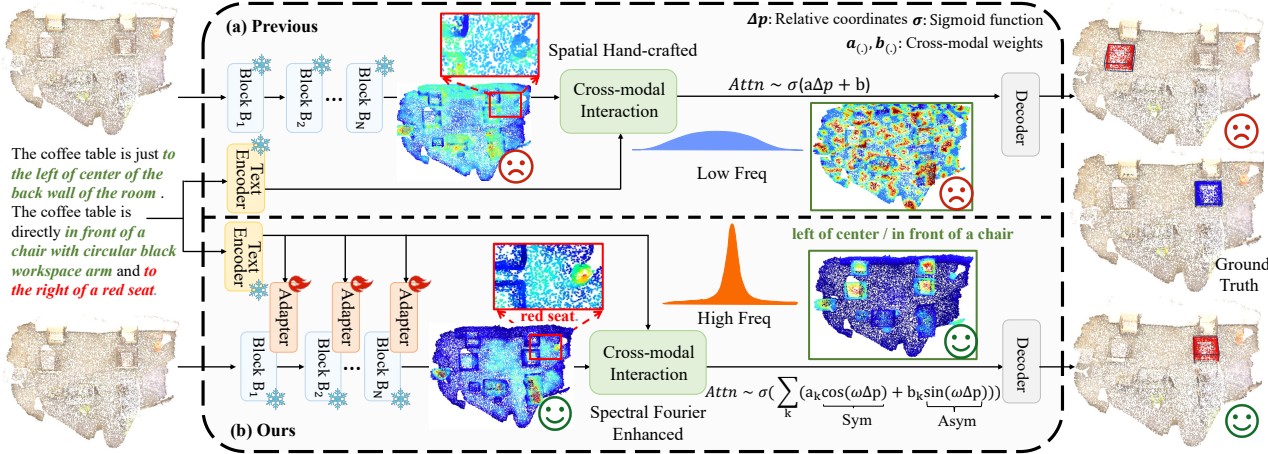

*Figure 1.* **(a)** Previous methods yield insufficient cross-modal representation. Relying on independent unimodal encoders causes semantic misalignment, while handcrafted priors under spectral bias produce low-frequency location attention $Attn$ and ambiguous grounding. **(b)** We introduce TFT for fine-grained semantic alignment and DSG to decompose relations into symmetric and antisymmetric components with learnable frequencies $\omega$. This helps to produce high-frequency peaks for precise grounding in 3D scenes.

Figure 1(a), directional details are often dominated or entangled by symmetric distance, resulting in ambiguous spatial matching. To address this, we propose decomposed spectral geometry module to enhance handcrafted priors. As shown in Figure 1(b), we extend random Fourier feature approximation of translation-invariant kernels by introducing learnable frequencies and explicitly decomposing pairwise relations into symmetric cosine and antisymmetric sine components. This design helps restore discriminative high-frequency geometric details essential for guiding language-conditioned location attention toward precise spatial modeling.

Specifically, Text-Aware Feature Tuning (TFT) comprises Task-Agnostic Semantic Calibration and Dynamic Gated Adapter. It injects linguistic context into the visual hierarchy to mitigate domain shift and enrich cross-modal semantic representation. Decomposed Spectral Geometry (DSG) is designed to capture complex object relations for precise geometric representation. It employs learnable Fourier frequencies to generate spectral features, which are then mapped to a dual-channel latent feature space via a multi-layer perceptron. This process decomposes pairwise relations into symmetric distance components and antisymmetric direction components, preserving essential low-frequency stability while recovering high-frequency details to refine object boundaries and reduce spatial ambiguity. Together, TFT and DSG respectively enhance semantic and geometric representation and improve performance on 3DREC and 3DRES. Our main contributions are summarized as follows:

1. We introduce Text-Aware Feature Tuning, the first parameter-efficient fine-tuning strategy designed for 3DVG. It adaptively injects linguistic context into the visual hierarchy to mitigate domain shift and enrich cross-modal representation for fine-grained semantic alignment.

2. We propose Decomposed Spectral Geometry, which decomposes pairwise relations into learnable symmetric and antisymmetric spectral components. By capturing high-frequency geometric details, DSG augments handcrafted priors to build a more robust geometric representation. This enables language-conditioned location attention to precisely retrieve complex spatial relations inherent in 3D scenes.

3. Extensive experiments on ScanRefer, Nr3D and Sr3D demonstrate the effectiveness of our proposed method. The promising results indicate that TFT and DSG provide robust cross-modal representation crucial for 3DREC and 3DRES.

## 2. Related Work

### 2.1. 3D Visual Grounding

Existing 3D visual grounding can be broadly categorized into two-stage and single-stage methods. Two-stage methods (Yuan et al., 2021; Hsu et al., 2023; Feng et al., 2021; Cai et al., 2022) utilize pre-trained 3D detection or segmentation models to generate candidate proposals, subsequently matching linguistic descriptions against these candidate proposals to identify the target. Single-stage methods (Guo et al., 2025; Wang et al., 2024; Luo et al., 2022; Wu et al., 2023) encode global point cloud for direct point-text matching, regressing bounding boxes or masks in an end-to-end approach. Recently, researchers have recognized the synergistic effects between 3DRES and 3DREC and are advancing toward unified frameworks. For example, 3DRefTR (Lin et al., 2023) integrates these tasks by appending a lightweight superpoint mask branch to 3DREC decoder for parallel prediction. MCLN (Qian et al., 2024b) employs a dual-branch decoder with an Adaptive Soft Alignment module for joint training. Unlike task-specific decoder designs, our work targets the core of cross-modal learning by refining

representation from both semantic and geometric perspectives, enabling 3DREC and 3DRES to mutually reinforce each other through a richly shared feature space.

## 2.2. Parameter-Efficient Fine-Tuning

Parameter-Efficient Fine-Tuning (PEFT) has been extensively adopted in Natural Language Processing (NLP) (Ding et al., 2023; Shi & Lipani, 2023; Sung et al., 2022) and 2D computer vision (Lian et al., 2022; Li et al., 2024; Chen et al., 2022a) to adapt large pre-trained models with low computational costs. Recently, the application of PEFT has extended to 3D point cloud tasks (Tang et al., 2024; Zha et al., 2023; Liang et al., 2025; Zhou et al., 2024). IDPT pioneers this direction by employing EdgeConv (Wang et al., 2019) to produce instance-aware prompts (Zha et al., 2023). DAPT further proposes a dynamic adapter that assigns dynamic scales based on token significance, seamlessly integrating with internal prompts to capture instance-specific features (Zhou et al., 2024). Transcending the spatial domain, PointGST introduces a spectral adapter to transfer inner confused tokens into the spectral domain, effectively eliminating feature correlations to achieve more precise tuning (Liang et al., 2025). However, the lack of cross-modal mechanisms limits the applicability of these methods in 3DVG. Inspired by DAPT, we propose text-aware feature tuning, which transcends uni-modal constraints by injecting linguistic information into the visual tuning process, achieving effective and precise cross-modal alignment.

## 2.3. Geometric Representation in 3D Scenes

3DVG requires precise reasoning about complex spatial relations. To capture geometric context, 3DVG-Transformer (Zhao et al., 2021) introduces a Coordinate-guided Contextual Aggregation module that injects sparse spatial proximity into attention, followed by a Multiplex Attention for cross-modal disambiguation. Similarly, 3DJCG (Cai et al., 2022) encodes relative Euclidean distances into relational embeddings, which are fused with self-attention to model object relations. ViL3D (Chen et al., 2022b) further advances this by designing a spatial self-attention that explicitly incorporates pairwise geometric distance and angles, modulating attention using language-conditioned location attention. In contrast, we enhance geometric representation in the spectral domain to alleviate spectral bias, enabling the model to capture high-frequency details and complex object relations.

# 3. Methodology

## 3.1. Preliminaries

**Translation-Invariant Kernels.** A kernel function $k : \mathbb{R}^d \times \mathbb{R}^d \to \mathbb{R}$ is translation-invariant if it depends only on the relative displacement between its inputs $x, y \in \mathbb{R}^d$, i.e.,

$k(x, y) = \kappa(x - y)$. Such kernels are widely adopted in spatial modeling because they characterize pairwise relations based on relative positions rather than absolute coordinates.

**Bochner's Theorem** A continuous and translation-invariant kernel $\kappa(\Delta)$ is positive-definite if and only if it admits the following Fourier representation: $\kappa(\Delta) = \int_{\mathbb{R}^d} e^{i\omega^\top \Delta} p(\omega) \, d\omega$, where $p(\omega)$ is a non-negative spectral density (Rudin, 2017). For real-valued kernels, this can simplify to the cosine form: $\kappa(\Delta) = \int_{\mathbb{R}^d} \cos(\omega^\top \Delta) \, p(\omega) \, d\omega$.

**Random Fourier Features (RFF).** RFF approximates $\kappa(\Delta)$ by Monte Carlo integration (Rahimi & Recht, 2007). Drawing $D$ frequencies $\{\omega_k\}_{k=1}^D \sim p(\omega)$, the feature map $z(x) \in \mathbb{R}^{2D}$ is defined as $z(x) = \frac{1}{\sqrt{D}} \left[ \cos(\omega_k^\top x), \sin(\omega_k^\top x) \right]_{k=1}^D$. According to the trigonometric identity, the inner product $z(x)^\top z(y)$ yields an unbiased estimator of the translation-invariant kernel:

$$z(x)^\top z(y) = \frac{1}{D} \sum_{k=1}^D \cos(\omega_k^\top (x - y)) \approx \kappa(x - y). \quad (1)$$

**Limitations: Symmetric Priors vs. Directional Geometry.** RFF in Eq.(1) is inherently restricted to even functions due to the symmetry of the cosine kernel, where $\kappa(\Delta) = \kappa(-\Delta)$. This restriction prevents the model from capturing non-commutative directional relations that are critical for 3DVG, such as whether $x$ is to the left or right of $y$. To address this, we employ the spectral decomposition implied by Euler's identity, which allows any real-valued translation-invariant kernel to be partitioned into a symmetric even component and an antisymmetric odd component. This parity decomposition serves as the principled foundation for our DSG, enabling a unified representation of both distance-based similarity and direction-dependent geometry.

**Learnable Fourier Frequencies.** Unlike RFF that typically sample frequencies $\{\omega_k\}_{k=1}^D$ from a fixed Gaussian distribution, we optimize the frequency matrix directly via backpropagation (Li et al., 2021), treating the spectral density $p(\omega)$ as a task-adaptive distribution. Consequently, the model can dynamically adjust its spectral bias to emphasize high-frequency components. This flexibility allows our DSG to adaptively tune its bandwidth across multiple spatial scales, facilitating precise modeling of fine-grained geometric structures that are important for 3DVG.

## 3.2. Overview

As shown in Figure 2, we employ a Set-Abstraction layer (Qi et al., 2017) and pre-trained 3DCLIP (Hegde et al., 2023) for 3D point cloud encoding, and use pre-trained RoBERTa (Liu et al., 2019) for text encoding. To achieve

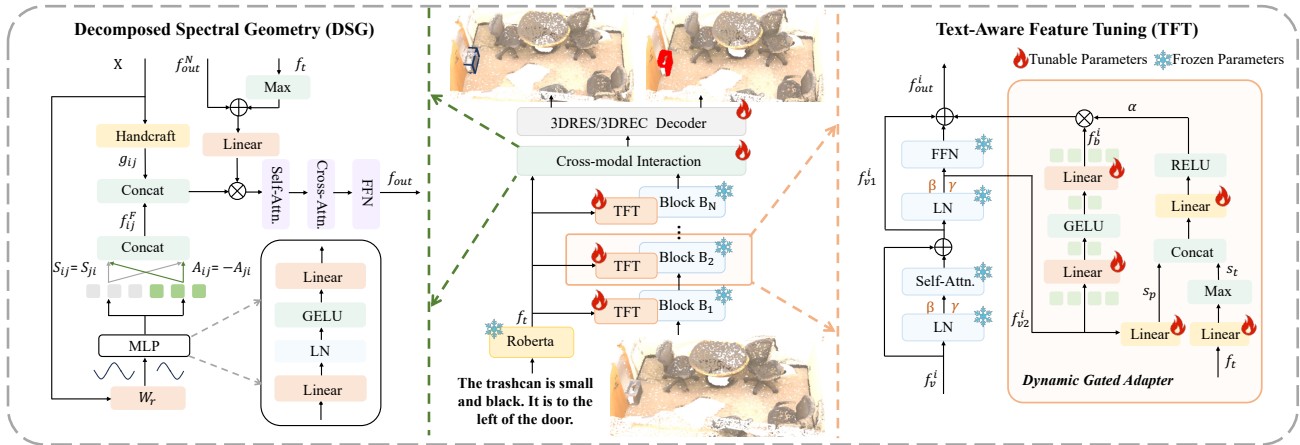

*Figure 2.* **Middle:** Overall architecture for joint 3DREC and 3DRES. **Left**: Coordinates are decomposed via learnable Fourier features into symmetric $S_{ij}$ and antisymmetric $A_{ij}$, capturing distance and direction to guide location attention toward precise spatial modeling. **Right:** TFT fuses visual importance $s_p$ and textual relevance $s_t$ to obtain a dynamic gate $\alpha$, adaptively refining cross-modal features.

fine-grained semantic alignment, we introduce Dynamic Gated Adapter that injects textual semantics into the visual encoder through a cross-modal gate. This enables the model to focus more on textual details in the early stage, generating aligned visual features $f_{out}^N \in \mathbb{R}^{N \times C}$ associated with the absolute coordinates $X \in \mathbb{R}^{N \times 3}$, where $N$ denotes the number of points in the point cloud and $C$ denotes the dimension. In the cross-modal fusion stage, we propose Decomposed Spectral Geometry. It decomposes pairwise relations into symmetric and antisymmetric spectral components, capturing high-frequency details to model more accurate object relations. Following MCLN (Qian et al., 2024b), we employ a multi-branch transformer decoder for 3DREC and 3DRES, optimizing the model with a joint multi-task objective.

### 3.3. Text-Aware Feature Tuning

To align cross-modal semantic features, we introduce Text-Aware Feature Tuning (TFT), adaptively adjusting pretrained features to meet the joint requirements of both 3DREC and 3DRES. TFT includes two components: Task-Agnostic Semantic Calibration and Dynamic Gated Adapter.

**Task-Agnostic Semantic Calibration.** We introduce Task-Agnostic Feature Calibration (TASC), which is designed to mitigate domain shift usually caused by directly transferring pretrained knowledge and implicitly modulate visual features to align with textual semantics under joint multimodal supervision. Given the point cloud features $f_v \in \mathbb{R}^{N \times C}$, TASC recalibrates feature distribution via (Lian et al., 2022):

$$\text{TASC}(x) = f_v \odot \gamma + \beta, \tag{2}$$

where $\odot$ denotes Hadamard product. The learnable $\gamma$ and $\beta$ are respectively initialized to 1 and 0 (Zhou et al., 2024), ensuring a stable optimization process that progressively adapts the visual space to the target domain in 3DVG.

**Dynamic Gated Adapter.** Building upon the implicitly calibrated features from TASC, we further propose Dynamic Gated Adapter (DGA) to explicitly enhance cross-modal semantic representation. By conditioning visual activations on textual semantics, it allows linguistic relevance to selectively modulate the saliency of specific visual features. Let $f_{v1}^i \in \mathbb{R}^{N \times C}$ denote the point cloud features produced by a self-attention layer and $f_{v2}^i \in \mathbb{R}^{N \times C}$ denote $f_{v1}^i$ after LayerNorm in the $i$-th Transformer block as shown in Figure 2. Let $f_t \in \mathbb{R}^{L \times C}$ denote the textual features extracted from RoBERTa and projected to the visual feature dimension $C$, where $L$ denotes the length of text tokens.

Specifically, we evaluate the contribution of each modality as formulated in Eq.(3). The visual importance score is obtained by the linear projection $W_p \in \mathbb{R}^{1 \times C}$ applied to the point cloud features. For the textual branch, we calculate importance scores for all $L$ tokens using the projection $W_t \in \mathbb{R}^{1 \times C}$ and apply max-pooling to aggregate the most salient global semantic information. These scores are then concatenated and fused through the projection $W_f \in \mathbb{R}^{1 \times 2}$ to generate a gate $\alpha$ as shown in Eq.(4), where $\|$ denotes concatenation and $\sigma$ is ReLU to ensure positive scaling.

$$s_p = W_p f_{v2}^i, \quad s_t = \max_{l \in \{1...L\}} (W_t f_t), \tag{3}$$

$$\alpha = \sigma(W_f[s_p \| s_t]), \tag{4}$$

Concurrently, the visual features $f_b^i$ are extracted using a bottleneck structure. The final features $f_{out}^i$ are obtained by Eq. (5), where $\varphi$ denotes GELU, $W_d \in \mathbb{R}^{r \times C}$ and $W_u \in \mathbb{R}^{C \times r}$ denote the down-projection and up-projection with rank $r$, respectively. Through the gating mechanism, $\alpha$ performs as a semantic filter that enhances visual features related to the text and suppresses irrelevant noise, producing more precise cross-modal semantic representation.

$$f_b^i = W_u \varphi(W_d f_{v2}^i), \quad f_{out}^i = f_{v1}^i + \alpha \odot f_b^i \tag{5}$$

## 3.4. Decomposed Spectral Geometry

Building upon the semantic enhancements from TFT, we introduce Decomposed Spectral Geometry (DSG) to enrich geometric representation for language-conditioned location attention. To represent spatial relations in 3D scenes, we utilize handcrafted pairwise geometric priors $g_{ij} \in \mathbb{R}^5$ following ViL3DRel (Chen et al., 2022b). Specifically, letting $x_i, x_j \in \mathbb{R}^3$ denote the absolute coordinates of points from $X$, we compute the relative displacement $\Delta p_{ij} = x_i - x_j$ alongside Euclidean distances and orientation angles. However, these handcrafted priors suffer from spectral bias that makes models prioritize low-frequency distance and smooth out high-frequency geometric details, which limits their capability to represent complex spatial relations. Thus, we employ learnable Random Fourier Features as summarized in Sec. 3.1. Distinct from typical methods that operate directly in the spatial domain, DSG projects spatial coordinates into the spectral domain and embeds the spectral features into a latent space to capture high-frequency details.

**Point-Level Spectral Decomposition.** We employ a learnable Fourier Frequencies strategy defined in Sec. 3.1 to construct a spectral feature space. For each point $x_i$, we compute a learnable random Fourier feature $r_i$:

$$r_i = \frac{1}{\sqrt{D}} \big[ \cos(W_r x_i) \ \| \ \sin(W_r x_i) \big] \in \mathbb{R}^D, \quad (6)$$

where $W_r \in \mathbb{R}^{D \times 3}$ is a learnable frequency matrix. As established in Eq.(1), the inner product $r_i^\top r_j$ approximates a translation-invariant kernel. However, this raw representation is constrained to modeling symmetric distance relations, failing to capture non-commutative directional ordering that is important for language-guided 3D understanding tasks.

To overcome this limitation, we move beyond static kernel approximation by introducing a task-adaptive geometric representation. Specifically, a nonlinear MLP projects the raw Fourier feature $r_i$ into an enriched spectral space and explicitly decomposes the output into two components:

$$\mathrm{MLP}(r_i) = \big[ u_i \ v_i \big] \in \mathbb{R}^{2d}, \quad u_i, v_i \in \mathbb{R}^d. \quad (7)$$

This decomposition establishes a dual-channel basis that enables the model to adaptively modulate spectral priors. Unlike fixed kernels in the Reproducing Kernel Hilbert Space (RKHS), this learned nonlinear mapping can represent the latent geometric structure of the 3D scene, providing the necessary flexibility to subsequently decompose pairwise relations into symmetric cosine and antisymmetric sine.

**Parity Decomposition in Dual Space.** Drawing on the Euler's identity parity decomposition principle discussed in Sec. 3.1, we reconstruct the pairwise geometry in the dual feature space using bilinear forms. As shown in Figure 2, we compute the symmetric $S_{ij}$ component and antisymmetric $A_{ij}$ component by element-wise operations:

$$S_{ij} = u_i \odot u_j + v_i \odot v_j \in \mathbb{R}^d, \quad (8)$$

$$A_{ij} = v_i \odot u_j - u_i \odot v_j \in \mathbb{R}^d. \quad (9)$$

For each spectral channel $k \in \{1, \ldots, D\}$, these components can be interpreted as bilinear forms of the joint feature $\phi_i = [u_i^\top, v_i^\top]^\top$

$$S_{ij}^{(k)} = \phi_i^\top M_{\mathrm{sym}}^{(k)} \phi_j, \quad A_{ij}^{(k)} = \phi_i^\top M_{\mathrm{as}}^{(k)} \phi_j, \quad (10)$$

where $M_{\mathrm{sym}}^{(k)} = \big[ \begin{smallmatrix} E_k & 0 \\ 0 & E_k \end{smallmatrix} \big]$ and $M_{\mathrm{as}}^{(k)} = \big[ \begin{smallmatrix} 0 & E_k \\ -E_k & 0 \end{smallmatrix} \big]$, $E_k = e_k e_k^\top \in \mathbb{R}^{d \times d}$. By Eq.(8) and Eq.(9), we extend standard scalar kernel approximations to a multi-channel spectral representation that captures complex spatial relations. The Fourier-enhanced priors $f_{ij}^F = [S_{ij} \| A_{ij}] \in \mathbb{R}^{2d}$ are formed by concatenating these spectral channels. Specifically, the symmetric component $S_{ij}$ characterizes order-invariant distance relations, while the antisymmetric $A_{ij}$ encodes order-sensitive directional information. This explicit decoupling enables DSG to approximate general translation-invariant functions and effectively align visual features with asymmetric linguistic prepositions such as "left" or "behind".

**Theoretical Insight: Relationship to Kernel Learning.** To understand the expressive power of our DSG module, consider the case where the nonlinear MLP is replaced by a linear mapping $\psi_i = Br_i$, where $B \in \mathbb{R}^{2d \times D}$. Under this simplification, the symmetric component in Eq. (8) reduces to $S_{ij}^{(k)} = r_i^\top (\mathbf{B}^\top M_{\mathrm{sym}}^{(k)} \mathbf{B}) r_j$, which is equivalent to a low-rank Fourier-domain factorization of a translation-invariant kernel. By introducing a nonlinear MLP, we relax this linear constraint, allowing the module to act as a task-adaptive geometric modeler that captures complex and non-stationary spatial structures beyond fixed stationary kernels. The advantage of this nonlinear parameterization over its linear counterpart is empirically validated by Ablation Study 4.4.

**Spectral-Enhanced Geometric Reasoning.** The final geometric prior $f_{ij}$ is formed by fusing handcrafted features with our Fourier-enhanced priors:

$$f_{ij} = \big[ g_{ij} \| f_{ij}^F \big] \in \mathbb{R}^{5+2d}. \quad (11)$$

This integrated representation is then used to modulate self-attention weights (Chen et al., 2022b). Crucially, the Fourier prior augments the cross-modal semantic representation by capturing high-frequency geometric nuances. This allows for fine-grained alignment between linguistic descriptions and 3D spatial structures, facilitating more accurate retrieval of complex object-to-object relations.

**Memory Efficiency Analysis.** We significantly reduce memory overhead compared to a naive implementation that

*Table 1.* Quantitative results of 3DREC on ScanRefer.

| Method | Unique | | Multiple | | Overall | |
|---|---|---|---|---|---|---|
| | 0.25 | 0.5 | 0.25 | 0.5 | **0.25** | **0.5** |
| ***Two-Stage Model*** | | | | | | |
| ScanRefer (Chen et al., 2020) | 76.33 | 53.51 | 32.73 | 21.11 | 41.19 | 27.40 |
| TGNN (Huang et al., 2021) | 68.61 | 56.80 | 29.84 | 23.18 | 37.37 | 29.70 |
| InstanceRefer (Yuan et al., 2021) | 77.45 | 66.83 | 31.27 | 24.77 | 40.23 | 32.93 |
| SAT (Yang et al., 2021) | 73.21 | 50.83 | 37.64 | 25.16 | 44.54 | 30.14 |
| 3DJCG (Cai et al., 2022) | 83.47 | 64.34 | 41.39 | 30.82 | 49.56 | 37.33 |
| BUTD-DETR (Jain et al., 2022) | 82.88 | 64.98 | 44.73 | 33.97 | 50.42 | 38.60 |
| EDA (Wu et al., 2023) | 85.76 | 68.57 | 49.13 | 37.64 | 54.59 | 42.26 |
| 3DRefTR (Lin et al., 2023) | 86.12 | 71.04 | 50.07 | 38.65 | 55.45 | 43.48 |
| VPP-Net (Shi et al., 2024) | 86.05 | 67.09 | 50.32 | 39.03 | 55.65 | 43.29 |
| MCLN (Qian et al., 2024b) | 86.61 | 71.04 | 51.13 | 40.56 | 56.43 | 45.11 |
| EDA+AugRefer (Wang et al., 2025) | 86.21 | 70.75 | 49.96 | 39.06 | 55.68 | 44.03 |
| UniSpace-3D (Zheng et al., 2025) | 86.72 | 70.07 | 50.56 | 39.89 | 56.04 | 43.95 |
| **DSGCR (Ours)** | **87.17** | **71.81** | **53.44** | **42.61** | **58.48** | **46.97** |
| ***Single-Stage Model*** | | | | | | |
| 3D-SPS (Luo et al., 2022) | 81.63 | 64.77 | 39.48 | 29.61 | 47.65 | 36.43 |
| EDA (Wu et al., 2023) | 86.40 | 69.42 | 49.03 | 37.93 | 53.83 | 41.70 |
| 3DRefTR (Lin et al., 2023) | 86.40 | 68.01 | 48.82 | 37.83 | 54.43 | 42.33 |
| MCLN (Qian et al., 2024b) | 84.43 | 68.36 | 49.72 | 38.41 | 54.30 | 42.64 |
| TSP3D (Guo et al., 2025) | 85.54 | **72.15** | 50.06 | **41.19** | 55.30 | **46.01** |
| **DSGCR (Ours)** | **86.47** | 71.74 | **50.48** | 38.48 | **55.85** | 43.44 |

*Table 2.* Quantitative results of 3DRES on ScanRefer.

| Method | Unique | | Multiple | | Overall | | **mIoU** |
|---|---|---|---|---|---|---|---|
| | 0.25 | 0.5 | 0.25 | 0.5 | **0.25** | **0.5** | |
| TGNN (Huang et al., 2021) | 69.3 | 57.8 | 31.2 | 26.6 | 38.6 | 32.7 | 28.8 |
| BUTD-DETR (Jain et al., 2022) | 76.63 | 63.30 | 38.01 | 29.70 | 45.53 | 36.22 | 35.47 |
| 3DRefTR (Lin et al., 2023) | 87.88 | 69.77 | 51.61 | 41.91 | 57.02 | 46.07 | 40.76 |
| X-RefSeg3D (Qian et al., 2024a) | - | - | - | - | 40.33 | 33.77 | 29.94 |
| 3D-STMN (Wu et al., 2024) | 89.3 | **84.0** | 46.2 | 29.2 | 54.6 | 39.8 | 39.5 |
| MCLN (Qian et al., 2024b) | 88.87 | 74.56 | 53.33 | 45.62 | 58.63 | 49.94 | 44.59 |
| **DSGCR (Ours)** | **89.43** | 76.25 | **54.96** | **47.68** | **60.11** | **51.95** | **45.68** |

explicitly constructs pairwise Fourier features $\cos(\omega^\top(x_i - x_j))$. This naive method requires instantiating an intermediate tensor of shape $(N, N, D)$, leading to a prohibitive space complexity of $O(N^2 D)$. In contrast, DSG decouples this process by performing Fourier projection and MLP modulation at the absolute coordinates, maintaining a complexity of $O(ND)$. We synthesize pairwise features only at the final stage with a reduced dimension $d$, effectively avoiding the need to generate massive intermediate tensors. Quantitatively, under our experimental settings with $N = 1024$ and $D = 32$, the naive approach demands $\sim 128\text{MB}$ ($1024^2 \times 32 \approx 3.36 \times 10^7$) of memory, whereas our DSG requires only $\sim 128\text{KB}$ ($1024 \times 32 \approx 3.28 \times 10^4$). This obvious memory reduction enables scalable training on point clouds without sacrificing geometric expressiveness.

# 4. Experiments

## 4.1. Datasets and Metrics

**Datasets.** To validate the effectiveness of our proposed method, we perform evaluations on three benchmarks: ScanRefer(Chen et al., 2020), Nr3D and Sr3D (Achlioptas et al., 2020). **ScanRefer** expanded its linguistic annotations across ScanNet, comprising 51,583 manually annotated sentences. **Nr3D** contains 41,503 human-annotated expressions and is designed to evaluate grounding performance under multi-instance. **Sr3D** comprises 83,572 template-based synthetic expressions across 1,273 scenes, classified into five spatial-relation categories to explore fine-grained geometric reasoning. We follow the official splits for training and validation.

**Metrics.** For the 3DREC task, we employ accuracy at specific IoU thresholds, denoted as Acc@$k$IoU where $k \in$ $\{0.25, 0.5\}$. For the 3DRES task, we report mIoU in addition to Acc@$k$IoU to evaluate segmentation performance.

## 4.2. Implementation Details

We implement our model using PyTorch (Paszke et al., 2019) and train it on two NVIDIA RTX 4090 GPUs. Input point clouds are subsampled to $N = 1024$ points and the model hidden dimension is set to $C = 384$. For the TFT module, the bottleneck rank is set to $r = 64$. For the DSG module, we use a frequency dimension of $D = 32$ and a spectral latent dimension of $d = 5$. Both the 3DCLIP visual encoder (Hegde et al., 2023) pretrained on ShapeNet (Chang et al., 2015) and the RoBERTa text encoder followed previous works are kept frozen during training. The learning rate and optimizer settings follow MCLN (Qian et al., 2024b).

## 4.3. Quantitative Analysis

**Performance on ScanRefer.** Tab. 1 and Tab. 2 report our results on ScanRefer following MCLN (Qian et al., 2024b), 3DRefTR (Lin et al., 2023) and EDA(Wu et al., 2023). For the 3DREC task, DSGCR demonstrates consistent superiority across both two-stage and single-stage settings. In the two-stage setting, we outperform MCLN by 2.05% in Acc@0.25 and 1.86% in Acc@0.5. Notably, on the Multiple split that requires instance disambiguation, we improve over MCLN by 2.31% in Acc@0.25. This improvement is primarily attributed to our DSG module, which explicitly decomposes pairwise geometric relations into symmetric and antisymmetric components. By effectively capturing both inter-object distances and directional orientations, DSG provides the critical spatial cues necessary for resolving ambiguity in cluttered and multi-object scenes. In the single-stage setting, while TSP3D tailored for 3DREC shows competitive Acc@0.5, DSGCR still attains the best Acc@0.25 across Unique, Multiple and Overall. For 3DRES task, DSGCR also leads other methods with an mIoU of 45.68%, surpassing MCLN by 1.09% and 3DRefTR by 4.92%. These results across both 3DREC and 3DRES validate that DSGCR strengthens cross-modal semantic representation, enabling the model to perform more precise predictions.

**Performance on Nr3D and Sr3D.** To verify the generalization capability of DSGCR, we report results on Nr3D and Sr3D in Tab. 3. On Sr3D, which is dominated by explicit

*Table 3.* Quantitative results of 3DREC on Nr3D and Sr3D.

| Method | Nr3D | | Sr3D | |
|---|---|---|---|---|
| | Hard | **Overall** | Hard | **Overall** |
| ReferIt3D (Achlioptas et al., 2020) | 27.9 | 35.6 | 31.5 | 40.8 |
| TGNN (Huang et al., 2021) | 30.6 | 37.3 | 36.9 | 45.0 |
| InstanceRefer (Yuan et al., 2021) | 31.8 | 38.8 | 40.5 | 48.0 |
| SAT (Yang et al., 2021) | 42.4 | 49.2 | 50.0 | 57.9 |
| 3D-SPS (Luo et al., 2022) | 45.1 | 51.5 | 65.4 | 62.6 |
| MVT (Huang et al., 2022) | 49.1 | 55.1 | 58.8 | 64.5 |
| 3DRefTR (Lin et al., 2023) | - | 52.6 | - | 68.5 |
| EDA (Wu et al., 2023) | 45.8 | 51.9 | 61.8 | 67.1 |
| MCLN (Qian et al., 2024b) | - | **59.8** | - | 68.4 |
| UniSpace-3D (Zheng et al., 2025) | 48.9 | 57.2 | **63.7** | 69.8 |
| **DSGCR (Ours)** | **52.0** | 57.9 | 63.1 | **70.0** |

spatial descriptions, DSGCR achieved an optimal performance of 70.0%. We also remain competitive on the challenging Nr3D, which features natural and diverse linguistic expressions. In Hard subset, DSGCR achieves 52.0%, outperforming UniSpace-3D 48.9% and EDA 45.8%. This suggests that DSGCR is effective in handling diverse spatial relations and can remain competitive across datasets.

### 4.4. Ablation Study

**Effectiveness of Proposed Modules.** We evaluate the contribution of each component in Tab. 4. The baseline model with a frozen visual encoder achieves 44.54% Acc@0.5 for 3DREC and 44.22% mIoU for 3DRES. Notably, full fine-tuning the backbone yields only a marginal improvement of 0.45% in Acc@0.5, suggesting that simple fine-tuning alone is insufficient to alleviate the cross-modal semantic gap. In contrast, incorporating the Text-aware Feature Tuning (TFT) module boosts performance, improving Acc@0.5 by 1.20% and mIoU by 1.15%. This confirms that early alignment of visual features with linguistic context is more effective than full fine-tuning. Similarly, introducing the Decomposed Spectral Geometry (DSG) module alone enhances mIoU by 1.13%, demonstrating its superior ability to capture high-frequency geometric details. When both modules are combined, DSGCR achieves the best results. It attains 46.97% Acc@0.5 for 3DREC and 45.68% mIoU for 3DRES, corresponding to gains of 2.43% in Acc@0.5 and 1.46% in mIoU over the baseline. The results indicate that the fine-grained semantic alignment provided by TFT and the high-frequency geometric refinement from DSG provide complementary benefits for 3D scene understanding.

**Ablation on TFT Components.** We further evaluate the individual contributions of Task-Agnostic Semantic Calibration (TASC) and the Dynamic Gated Adapter (DGA) as shown in Tab. 5. While TASC independently improves the baseline by aligning the global feature distribution with the requirements of the 3DVG task, its performance gains are relatively constrained. The DGA module enriches cross-modal representation by explicitly injecting fine-grained

*Table 4.* Ablation study on components and strategies. Full FT denotes full fine-tuning. Linear DSG represents the degenerate case using linear mapping without the nonlinear MLP.

| Method | FT | Components | | 3D REC | | 3D RES | | |
|---|---|---|---|---|---|---|---|---|
| | | TFT | DSG | 0.25 | 0.5 | 0.25 | 0.5 | mIoU |
| Baseline | | | | 56.29 | 44.54 | 58.00 | 50.39 | 44.22 |
| Full FT | ✓ | | | 56.90 | 44.99 | 59.05 | 51.05 | 44.92 |
| DSG (Linear) | | | ✓ | 57.46 | 45.70 | 58.97 | 50.62 | 44.85 |
| | | ✓ | | 57.47 | 45.74 | 59.62 | 51.54 | 45.37 |
| DSGCR (Ours) | | | ✓ | 57.91 | 45.49 | 59.68 | 51.46 | 45.35 |
| | | ✓ | ✓ | **58.48** | **46.97** | **60.11** | **51.95** | **45.68** |

linguistic context into the visual hierarchy by a dynamic gating mechanism. This is particularly effective for segmentation, where it achieves 45.11% mIoU. The combination of both components achieves optimal performance, with Acc@0.5 increasing by 0.98% over TASC alone. This indicates that TASC provides a stable feature calibration and DGA effectively modulates visual saliency based on linguistic semantics to achieve better cross-modal alignment.

**Ablation on DSG Components.** To validate the necessity of the parity decomposition in modeling pairwise geometric relations, we evaluate symmetric and antisymmetric spectral components in Tab. 6. The symmetric (Sys) component alone, which primarily models distance-aware relations, provides a 0.74% improvement in 3DREC Acc@0.5 over the baseline. The antisymmetric (A-Sys) component further enhances performance, leading to a 0.17% improvement in 3DRES mIoU over the symmetric component. The complete DSG module that integrates both components outperforms the performance of either component alone. This result highlights that capturing complex spatial dependencies necessitates unifying the representation of distance-based similarity with the encoding of directional relations, which are critical for resolving spatial ambiguities in 3D visual grounding.

**Ablation on DSG Hyperparameters.** We further evaluate the robustness of DSG to its two key hyperparameters: the frequency dimension $D$ and the spectral latent dimension $d$. As shown in Tab. 7, the default setting is $(D, d) = (32, 5)$. When fixing $D = 32$, reducing $d$ to 3 leads to clear degradation, suggesting insufficient latent capacity for modeling decomposed spectral relations. Although increasing $d$ to 8 leads to a slight improvement in the mIoU of 3DRES, it degrades the performance on 3DREC. Therefore, $d = 5$ achieves the best overall trade-off across both tasks. With $d = 5$ fixed, $D = 32$ consistently performs best, while both smaller and larger values of $D$ degrade performance.

**Linear vs. Nonlinear Mapping in DSG.** To verify the theoretical insight in Sec. 3.4, we compare the proposed nonlinear DSG with linear DSG in Tab. 4, which reduces to a shift-invariant kernel factorization. While the linear DSG provides a baseline improvement, the nonlinear MLP further improves performance, particularly in 3DREC where

| Description | GT | EDA | MCLN | Ours |
|---|---|---|---|---|

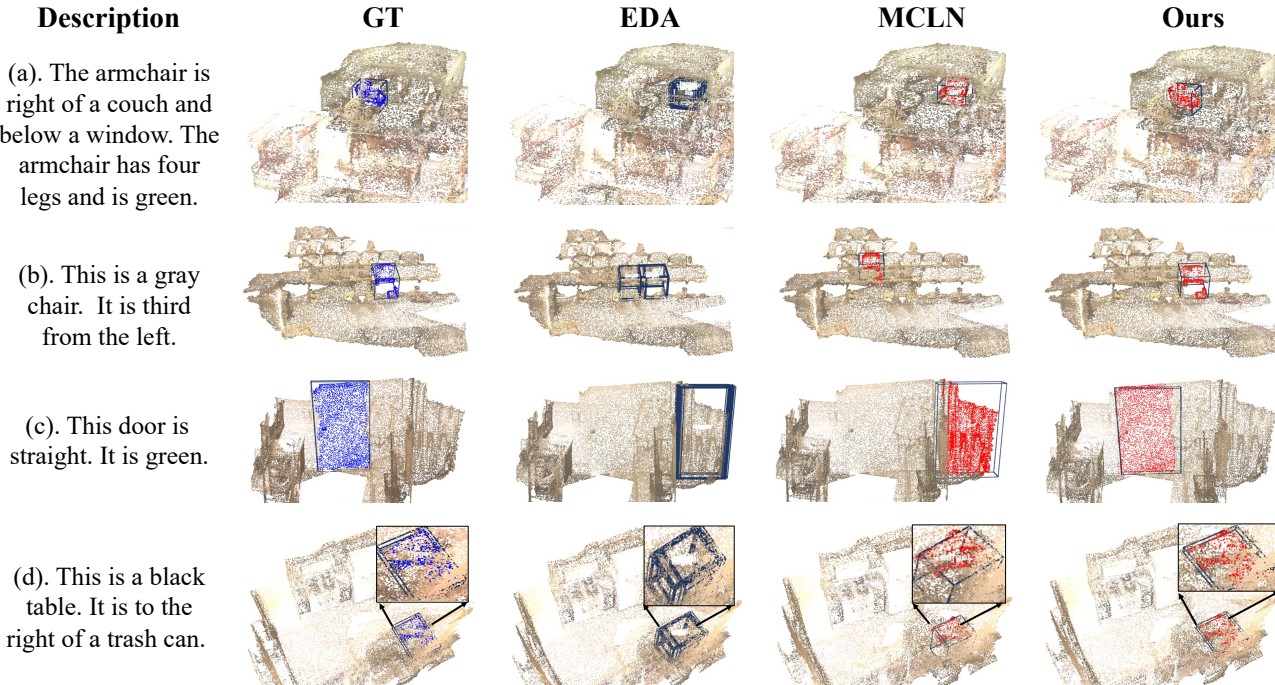

(a). The armchair is right of a couch and below a window. The armchair has four legs and is green.

(b). This is a gray chair. It is third from the left.

(c). This door is straight. It is green.

(d). This is a black table. It is to the right of a trash can.

*Figure 3.* Qualitative results from EDA, MCLN and our DSGCR.

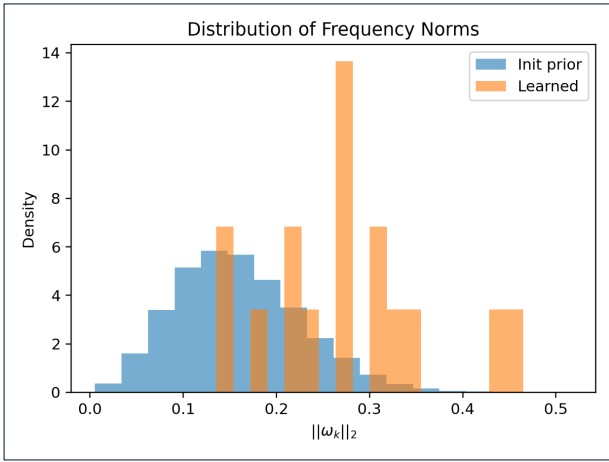

*Figure 4.* Learned frequency distribution of DSG

*Table 5.* Ablation study of the TFT module.

| TFT | | 3D REC | | 3D RES | | |
|---|---|---|---|---|---|---|
| TASC | DGA | 0.25 | 0.5 | 0.25 | 0.5 | mIoU |
| ✓ | | 57.18 | 44.76 | 59.38 | 50.86 | 44.84 |
| | ✓ | 57.32 | 44.74 | **59.66** | 51.19 | 45.11 |
| ✓ | ✓ | **57.47** | **45.74** | 59.62 | **51.54** | **45.37** |

*Table 6.* Ablation study of the DSG module.

| DSG | | 3D REC | | 3D RES | | |
|---|---|---|---|---|---|---|
| Sys | A-Sys | 0.25 | 0.5 | 0.25 | 0.5 | mIoU |
| ✓ | | 57.25 | 45.28 | 58.77 | 50.80 | 44.68 |
| | ✓ | 57.39 | 44.58 | 59.32 | 51.01 | 44.85 |
| ✓ | ✓ | **57.91** | **45.49** | **59.68** | **51.46** | **45.35** |

the Acc@0.5 increases from 45.70% to 46.97%. This confirms that the nonlinear parameterization allows the model to capture complex, non-stationary spatial structures that exceed the capacity of fixed stationary kernels, providing the necessary flexibility for precise 3DREC and 3DRES.

### 4.5. Qualitative Analysis

Figure 3 presents qualitative comparisons on ScanRefer among the Ground Truth, EDA, MCLN and our DSGCR. Existing methods often fail at complex spatial reasoning and fine-grained semantic alignment. As illustrated in (a-c), EDA misinterprets ordinal expressions like "third from the left", while MCLN has difficulty parsing directional cues

and attribute-specific descriptions in cluttered 3D scenes. In contrast, DSGCR achieves more precise grounding. Leveraging the TFT module to inject linguistic context and mitigate domain shift, our model correctly resolves semantic attributes such as "the straight green door". Meanwhile, the DSG module captures high-frequency geometric details and direction-aware relations beyond handcrafted priors, enabling reliable localization of spatial concepts such as "left" and "right". Crucially, as illustrated in (d), the recovery of these intricate spectral components allows DSGCR to delineate sharp object boundaries and maintain robust cross-task consistency between 3DREC and 3DRES outputs. Importantly, as shown in (d), enhanced cross-modal semantic representation allows DSGCR to produce consis-

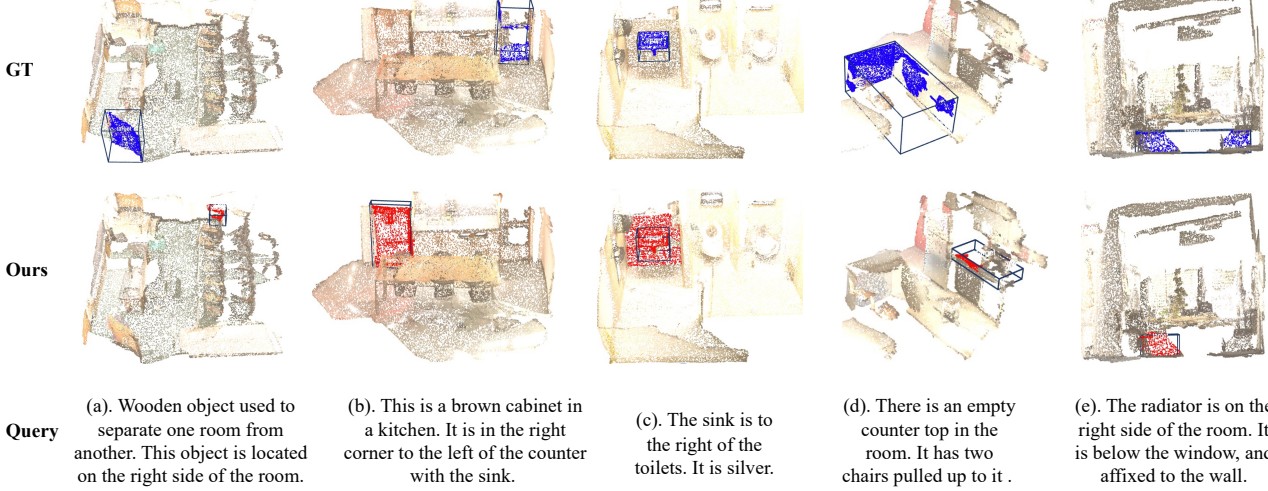

|  | GT |
|---|---|
|  | Ours |
| Query | (a). Wooden object used to separate one room from another. This object is located on the right side of the room. | (b). This is a brown cabinet in a kitchen. It is in the right corner to the left of the counter with the sink. | (c). The sink is to the right of the toilets. It is silver. | (d). There is an empty counter top in the room. It has two chairs pulled up to it. | (e). The radiator is on the right side of the room. It is below the window, and affixed to the wall. |

*Figure 5.* Failure case analysis of our DSGCR.

*Table 7.* Ablation study of the DSG hyperparameters.

| Hyperparameters | | 3D REC | | 3D RES | | mIoU |
|---|---|---|---|---|---|---|
| $D$ | $d$ | 0.25 | 0.5 | 0.25 | 0.5 | |
| 32 | 5 | **58.48** | **46.97** | 60.11 | **51.95** | 45.68 |
| 32 | 3 | 56.43 | 44.78 | 58.40 | 50.63 | 44.45 |
| 32 | 8 | 57.72 | 45.98 | **60.14** | 51.82 | **45.76** |
| 16 | 5 | 57.51 | 44.96 | 59.35 | 51.10 | 45.05 |
| 64 | 5 | 57.17 | 44.39 | 58.38 | 50.06 | 44.22 |

tent predictions between 3DREC boxes and 3DRES masks.

To directly analyze spectral behavior, we also compare the norms of the initialized frequencies and the learned frequencies in Fig. 4. The results show that DSG adapts its Fourier basis to incorporate higher-frequency components.

## 5. Limitations

**Failure Cases.** To better analyze the limitations of DSGCR, we visualize representative failure cases in Fig. 5. (a)-(b) show that the model struggles to select correct viewpoint and grounds a target to an incorrect object under ambiguous descriptions. (c) reveals inconsistency between the 3DREC box and the 3DRES mask, suggesting that the two task heads do not always share aligned intermediate representation. (d) shows that DSGCR tends to focus on the primary spatial cue while neglecting secondary relational constraints, indicating that our current spectral geometry modeling mainly captures pairwise relations independently rather than reasoning over relational chains. Finally, (e) reflects inaccurate segmentation boundaries and unreliable box predictions when objects are occluded or represented by only a sparse set of points.

**Generalization and broader geometric modeling.** Although DSG is not a purely handcrafted design, it still introduces inductive biases through symmetric and antisymmet-ric decomposition, motivated by directional and relational language common in 3D scene understanding. These inductive biases may affect its generalization to larger-scale and more diverse data, and thus its robustness in broader scenarios remains to be fully validated. Moreover, our current comparisons mainly focus on widely adopted handcrafted-prior methods in 3DVG. Future work could include broader comparisons with learnable geometric modeling and explore how such geometric modeling can be integrated into stronger pretrained or MLLM-based 3D grounding models.

## 6. Conclusion

In this study, we propose DSGCR to enhance 3D cross-modal semantic representation by incorporating linguistic semantics and geometric relations, improving joint 3DREC and 3DRES performance. We introduce Text-Aware Feature Tuning, which injects linguistic context into the visual hierarchy by a dynamic gating mechanism to mitigate domain shift and enable the model to capture cross-modal semantics precisely. To address complex spatial ambiguities, we propose Decomposed Spectral Geometry. It leverages learnable Fourier frequencies and maps random Fourier features to a spectral latent space to decompose pairwise relations into symmetric and antisymmetric spectral components, capturing high-frequency geometric details and direction-aware relations for better spatial reasoning. Extensive experiments conducted on the ScanRefer, Nr3D and Sr3D demonstrate the effectiveness of our method in enhancing cross-modal semantic representation for 3D scene understanding.

## Acknowledgements

This work was supported in part by the Joint Funds of the National Natural Science Foundation of China

(U22B2054), the National Natural Science Foundation of China (62076192, 62276199, 62431020 and 62276201), the 111 Project, the Program for Cheung Kong Scholars and Innovative Research Team in University (IRT 15R53), the Science and Technology Innovation Project from the Chinese Ministry of Education and the National Key Laboratory of Human-Machine Hybrid Augmented Intelligence, Xi'an Jiaotong University (HMHAI-202404 and HMHAI-202405).

## Impact Statement

This work improves language-guided 3D scene understanding by enhancing cross-modal alignment and spatial reasoning for 3D visual grounding. It may benefit embodied AI, robotics, augmented reality and assistive systems that require accurate interaction with complex 3D environments. Potential risks include inaccurate grounding and privacy concerns related to indoor scene data, which should be addressed through careful evaluation and human oversight.

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
