# OpenReview forum: "DSGCR: Decomposed Spectral Geometry-Aware Cross-Modal Semantic Representation for 3D Visual Grounding"
_ICML.cc/2026/Conference — ICML 2026 regular_

### Official Review · Reviewer_rRAA · 2026-02-27

**Soundness:** 3
**Presentation:** 3
**Significance:** 3
**Originality:** 3
**Overall Recommendation:** 4
**Confidence:** 3

**Summary:**

This paper studies 3D visual grounding (3DREC and 3DRES) and identifies two key bottlenecks: weak vision-language alignment due to independently transferred unimodal pretraining, and limited geometric reasoning from handcrafted spatial priors that under-represent high-frequency details and directional relations. To address these, it proposes DSGCR with (1) Text-aware Feature Tuning (TFT), a parameter-efficient tuning method that injects textual context into the visual hierarchy via feature calibration and a dynamic gated adapter for earlier, finer-grained cross-modal alignment, and (2) Decomposed Spectral Geometry (DSG), which uses learnable Fourier features and explicitly decomposes pairwise relations into symmetric (distance-like) and antisymmetric (direction-aware) spectral components to enhance spatial modeling. Experiments on ScanRefer, Nr3D, and Sr3D show state-of-the-art or competitive gains, and ablations confirm the complementary benefits of TFT and DSG.

**Compliance With Llm Reviewing Policy:**

Affirmed.

**Key Questions For Authors:**

See Weakness

**Limitations:**

See Weakness

**Strengths And Weaknesses:**

Strength：
1. TFT injects text information into the visual hierarchy via a dynamic gated adapter, providing parameter-efficient tuning that reduces domain shift without fully fine-tuning large backbones.
2. DSG uses learnable Fourier features and explicitly decomposes pairwise relations into symmetric (distance-like) and antisymmetric (direction-aware) components, better matching language involving spatial directions.
3. Results on ScanRefer/Nr3D/Sr3D and ablations (TFT vs DSG, symmetric vs antisymmetric, linear vs nonlinear DSG) support that each component contributes and the combination performs best.

Weakness：
1. The paper does not thoroughly study how hyperparameters and design choices (e.g., Fourier dimension \(D\), latent dimension \(d\), MLP structure, pooling/gating strategy) affect robustness and performance.
2. The claim that DSG mitigates spectral bias is supported mainly by downstream metrics rather than direct analysis (e.g., learned frequency distributions or targeted high-frequency geometric stress tests).

---

> ### Author Rebuttal · Authors · 2026-03-31
>
> Dear Reviewer rRAA,
>
> We thank the reviewer for these constructive suggestions, which help us to polish the paper. We agree that the paper would benefit from a more rigorous discussion on hyperparameter robustness, as well as a direct analysis of the learned frequency. In response, we provide an expanded analysis on the key DSG hyperparameters D and d, and direct frequency analysis of the learned Fourier basis. We will add this expanded analysis to the appendix and clarify it in the revised paper.
>
> # Weakness 1
>
> **Sensitivity analysis of the DSG hyperparameters on ScanRefer val. The default setting is $(D,d)=(32,5)$**
> | Fourier Dim ($D$) | Latent Dim ($d$) | 3DREC@0.25 | 3DREC@0.5 | 3DRES@0.25 | 3DRES@0.5 | mIoU |
> | :---: | :---: | :---: | :---: | :---: | :---: | :---: |
> | **32**| **5** | **58.48** | **46.97** | 60.11 | **51.95** | 45.68 |
> | 32 | 3 | 56.43 | 44.78 | 58.40 | 50.63 | 44.45 |
> | 32 | 8 | 57.72 | 45.98 | **60.14** | 51.82 | **45.76** |
> | 16 | 5 | 57.51 | 44.96 | 59.35 | 51.10 | 45.05 |
> | 64 | 5 | 57.17 | 44.39 | 58.38 | 50.06 | 44.22 |
>
> We further evaluate the robustness of DSG to its two key hyperparameters: the Fourier dimension D and the latent spectral dimension d. When fixing D=32, reducing d to 3 leads to clear degradation, suggesting insufficient latent capacity for modeling decomposed spectral relations. Increasing d to 8 slightly improves RES mIoU but reduces REC accuracy, while d=5 gives the best overall trade-off across the two tasks. When fixing d=5, D=32 consistently performs best, whereas both a smaller basis (D=16) and an overly large basis (D=64) degrade performance. This suggests that DSG does not depend on excessively large hyperparameter settings.
>
> # Weakness 2
>
> To direct analysis we now provide a frequency analysis of the learned Fourier basis used in DSG (Now available in https://anonymous.4open.science/r/EE2DXX2002/Figure/freq_analysis.png). Specifically, we compare the $L_2$ norms of the initialized frequency and the learned frequency after training. The empirical results demonstrate a clear and consistent shift toward higher frequencies. (i)The histogram shows more mass concentrated at larger values compared to the initialization prior. (ii) The Cumulative Distribution Function (CDF) of the learned norms is distinctly right-shifted across all quantiles, and (iii) the sorted learned norms lie consistently above the initialization mean.
>
> Since larger $\|\|\omega_k\|\|_2$ correspond to higher spatial frequencies in the Fourier basis, this provides direct evidence that DSG does not simply remain near the low-frequency initialization prior. Instead, training reallocates spectral mass toward higher-frequency components, which is consistent with improved sensitivity to fine-grained geometry and directional relations.
>
> Best regards,
>
> Wishing you a wonderful day!
>
> Authors

---

> > ### Author Rebuttal · Reviewer_rRAA · 2026-04-03
> >
> > I recommend Weak Accept since the paper presents a solid 3D visual grounding method with innovative TFT and DSG designs, strong experimental results, and the authors have thoroughly addressed my concerns on hyperparameter sensitivity and frequency analysis in the rebuttal.

---

> > > ### Author Response · Authors · 2026-04-03
> > >
> > > Dear Reviewer rRAA,
> > >
> > > Thank you very much for your positive assessment and for your thoughtful feedback. We sincerely appreciate your recognition of the innovative aspects of our TFT and DSG designs, as well as your acknowledgement that **our rebuttal thoroughly addressed your concerns on hyperparameter sensitivity and frequency analysis.**
> > >
> > > Best regards,
> > >
> > > Authors

---

### Official Review · Reviewer_NiqF · 2026-03-07

**Soundness:** 3
**Presentation:** 3
**Significance:** 2
**Originality:** 2
**Overall Recommendation:** 4
**Confidence:** 3

**Summary:**

This paper proposes DSGCR, a unified framework addressing cross-modal misalignment and spectral bias in 3D visual grounding (3DVG). It introduces two core modules. First, Text-aware Feature Tuning (TFT) injects linguistic context into the visual encoder via dynamic gating to achieve fine-grained semantic alignment. Second, Decomposed Spectral Geometry (DSG) maps learnable Fourier features into a latent space, decoupling spatial relations into symmetric (distance) and antisymmetric (direction) components to capture high-frequency geometric details. DSGCR achieves state-of-the-art performance on 3DREC and 3DRES tasks across multiple benchmarks.

**Compliance With Llm Reviewing Policy:**

Affirmed.

**Final Justification:**

The authors' rebuttal has solved my concerns, and I keep my original positive rating based on the paper's strengths.

**Key Questions For Authors:**

See weaknesses.

**Limitations:**

yes

**Strengths And Weaknesses:**

**Strengths**

- To address the complex spatial relationship reasoning problem in 3D visual grounding, the paper proposes the Decomposed Spectral Geometry (DSG) module. By utilizing learnable Fourier features, the authors decouple pairwise geometric relations at the feature level into symmetric (distance) and antisymmetric (direction) components. This approach provides a sound mathematical framework and solution for mitigating the model's spectral bias and handling non-commutative directional relations (e.g., distinguishing "left" from "right").

- The paper conducts thorough testing on multiple benchmark datasets, including ScanRefer, Nr3D, and Sr3D, demonstrating competitive performance on both 3DREC and 3DRES tasks. Furthermore, the authors optimize the engineering implementation of the module, reducing spatial complexity by decoupling the feature synthesis process, which holds practical application value for processing large-scale point cloud data.

**Weaknesses**

- The authors do not provide a visualization or quantitative analysis of the spectral distribution of the learned frequency matrix $\omega$. Did the model genuinely learn higher-frequency components, or did it achieve better fitting capability simply due to the addition of a nonlinear MLP (as shown in Table 4)? Additionally, it is recommended that the authors visualize the attention maps modulated by the symmetric component $S_{ij}$ and the antisymmetric component $A_{ij}$ to intuitively demonstrate that the module successfully decouples "distance" and "direction-aware" relations.

- It is unclear whether the "Baseline" in Table 4 includes the handcrafted priors $g_{ij}$. If it does, the improvement of DSG indeed comes from the proposed module; if not, the performance gain might simply result from the injection of spatial information rather than the superiority of DSG's parity decomposition. The authors should supplement ablation studies to verify the respective contributions of $g_{ij}$ and the DSG priors. Furthermore, if DSG can perfectly model complex spatial relations as theoretically claimed, why does it still rely on the handcrafted priors $g_{ij}$? This presents a minor logical conflict.

- The Task-Agnostic Semantic Calibration (essentially an affine transformation) and Dynamic Gated Adapter (essentially a bottleneck layer with cross-modal gating) utilized in TFT are highly standard and mature techniques in Parameter-Efficient Fine-Tuning (PEFT) and large vision-language models (e.g., GLIP, MDETR, or earlier multi-modal fusion works). While applying a simple combination of these two techniques to the 3DVG domain is effective (as shown in Table 5), it lacks a significant methodological innovation.

---

> ### Author Rebuttal · Authors · 2026-03-31
>
> Dear Reviewer NiqF,
>
> We have carefully considered all your questions and will address them one by one below.
> # Weakness1
> We thank the reviewer for this constructive suggestion. Regarding whether the model genuinely learns higher-frequency components, we now provide a direct frequency analysis of the learned Fourier basis (Now available in https://anonymous.4open.science/r/EE2DXX2002/Figure/freq_analysis.png). The empirical results demonstrate a consistent and clear shift toward higher frequencies after training. The histogram shows more spectral mass concentrated at larger norm values compared to initialization. The Cumulative Distribution Function (CDF) of learned norms is distinctly right-shifted across all quantiles, and the sorted learned norms lie consistently above the initialization mean. Since larger norms correspond to higher spatial frequencies, this provides direct evidence that DSG reallocates part of its spectral mass toward higher-frequency components. This also helps answer about whether the gain comes only from the nonlinear MLP. Moreover, our current ablation already shows that even the linear DSG variant improves over the baseline, while the nonlinear mapping further improves performance, indicating that the benefit comes from both the learned spectral representation and the nonlinear refinement, rather than from the MLP alone.
>
> In addition, following the reviewer’s suggestion, we now visualize the attention maps modulated by the antisymmetric and symmetric components(Now available in https://anonymous.4open.science/r/EE2DXX2002/Figure/visual_attn.png). Qualitatively, the antisymmetric term produces a spatially asymmetric attention pattern, where different parts of the scene receive clearly distinct attention weights according to their relative spatial orientation. This is consistent with the order-sensitive nature of the antisymmetric component and aligns well with the directional prepositions in the query such as "left of", "in front of" and "to the right of". By contrast, the symmetric is invariant to the ordering of the point pair and cannot by itself distinguish directional relations. Instead, it is better interpreted as an order-invariant, distance-like affinity cue, which provides compact local support over spatially compatible neighborhoods.The contrast between these two maps provides intuitive evidence that DSG decouples direction-aware and distance-like geometric reasoning.
> # Weakness2
> We thank the reviewer for pointing out this ambiguity. We apologize for the insufficient description in the paper. The Baseline in Table 4 does include the handcrafted priors, meaning that DSG is applied on top of these priors and its improvement can be attributed to the contribution of the proposed DSG. Regarding why DSG still uses handcrafted priors,we would like to respectfully note that the two are complementary. Specifically, handcrafted priors encoding Euclidean distances and projected angles provide stable, metric-level spatial measurements that are effective for capturing coarse proximity information, while DSG focuses on recovering high-frequency directional relations such as left/right ordering that are non-commutative and difficult to represent with scalar distance metrics alone. The integration in Eq.(11) thus yields a richer and more complete geometric representation, and we will revise the manuscript to make both the baseline setting and this complementary relationship more explicit.
> # Weakness3
> We appreciate this comment and agree that the individual building blocks in TFT are lightweight and related to established PEFT modules. We would like to respectfully note that the contribution of TFT lies less in the individual components themselves and more in how they are adapted to the cross-modal setting of 3D visual grounding. Unlike prior PEFT methods for 3D point clouds, which focus exclusively on unimodal adaptation, TFT is specifically designed to inject textual semantics into the 3D visual hierarchy, thereby reducing the visual-language domain gap in a parameter-efficient manner. This setting differs from prior PEFT methods for 3D point clouds, which mainly focus on unimodal point-cloud understanding without explicit cross-modal adaptation. During development, we also evaluated other alternatives such as text-only gating and mean pooling for text performed worse than the current joint point-text gate with max pooling. This comparison motivated our final TFT formulation. We also agree that exploring more sophisticated cross-modal adapter designs for 3D scenes is a valuable direction for future work.
>
> Best regards,
>
> Wishing you a wonderful day!
>
> Authors

---

> > ### Author Rebuttal · Reviewer_NiqF · 2026-04-03
> >
> > Thanks for the rebuttal and I will keep my positive rating.

---

> > > ### Author Response · Authors · 2026-04-03
> > >
> > > Dear Reviewer NiqF,
> > >
> > > Thank you very much for your careful review and for your positive assessment. We sincerely appreciate your acknowledgement that our rebuttal have adequately addressed your concerns.
> > >
> > > Best regards,
> > >
> > > Authors

---

### Official Review · Reviewer_oawD · 2026-03-13

**Soundness:** 3
**Presentation:** 3
**Significance:** 2
**Originality:** 2
**Overall Recommendation:** 4
**Confidence:** 4

**Summary:**

This paper proposes **DSGCR** for 3D visual grounding. The paper argues that existing methods suffer from weak cross-modal alignment and limited geometric reasoning. To address this, the authors introduce Text-aware Feature Tuning (TFT) to inject linguistic context into the visual hierarchy, and Decomposed Spectral Geometry (DSG) to model pairwise relations with learnable spectral components. Overall, the authors assess a central concept: that better early cross-modal tuning and frequency-aware geometric modeling can substantially improve fine-grained 3D grounding. Experiments on ScanRefer, Nr3D, and Sr3D show strong results.

**Compliance With Llm Reviewing Policy:**

Affirmed.

**Final Justification:**

The rebuttal addressed my main concerns, and I maintain my positive score.

**Key Questions For Authors:**

1. Can the authors clarify how much of the gain from TFT reflects stronger cross-modal alignment specifically, rather than generic adapter-based feature tuning on top of frozen 3DCLIP features?

2. Can the authors provide stronger justification that the proposed spectral decomposition in DSG is preferable to other learnable geometric modeling alternatives, beyond standard handcrafted priors?

3. Could the empirical positioning be stated more carefully, given that the method is strong but not uniformly best across all benchmark settings, e.g., Nr3D overall?

4. Have the authors evaluated whether TFT and DSG generalize to other 3D visual encoders beyond the current Set-Abstraction + 3DCLIP pipeline?

5. Can the authors provide a more substantive discussion of limitations or failure cases?

**Limitations:**

No. The discussion of limitations is limited. The paper would benefit from a clearer discussion of failure cases, robustness, and practical scope.

**Strengths And Weaknesses:**

***Strengths:***

The paper addresses a meaningful problem in 3D visual grounding, namely the joint need for fine-grained semantic alignment and precise spatial reasoning. The proposed design includes the TFT to improve early cross-modal alignment, and the DSG to enrich geometric priors with learnable spectral decomposition. The empirical evaluation shows strong results on ScanRefer, competitive performance on Nr3D/Sr3D, and ablations that support both modules as well as the nonlinear spectral design.

***Weaknesses:***

1. The claimed benefit of TFT is early cross-modal alignment, but the evidence remains somewhat indirect; it is not fully clear how much it improves semantic alignment beyond acting as a lightweight PEFT-style adapter on top of frozen 3DCLIP features.

2. The geometric contribution of DSG is interesting, but the comparison is still mainly against handcrafted-prior baselines within the same framework. It would be more convincing to see stronger evidence that the spectral decomposition is preferable to other learnable geometric modeling alternatives, not just to standard low-frequency priors.

3. The results on Nr3D/Sr3D are good but not uniformly dominant across all settings; for example, the method is not best on Nr3D overall in Table 3, so some empirical positioning should be stated a bit more carefully.

4. The generality of the proposed method is not fully validated, since the experiments are mainly built on a specific 3D visual encoder pipeline (i.e., Set-Abstraction + 3DCLIP). It remains unclear whether TFT and DSG would provide similar gains on other 3D backbones.

5. The paper provides limited discussion of the method’s limitations and failure cases, making it harder to assess its practical scope and potential weaknesses.

---

> ### Author Rebuttal · Authors · 2026-03-31
>
> Dear Reviewer oawD,
>
> # Weakness1 and Question1
> We sincerely thank the reviewer for this insightful question. To isolate the role of cross-modal alignment, we refer to the TFT ablation in Table 5. TFT has two components with different functions: TASC mainly serves as task-agnostic feature calibration, while DGA is the alignment-specific component because its gate is jointly determined by 3D visual features and textual semantics. As shown in Table 5, DGA alone improves the baseline from 56.29 to 57.32 on 3DREC Acc@0.25 and increases 3DRES mIoU from 44.22 to 45.11, showing that explicit text-conditioned modulation is beneficial beyond generic feature tuning. When combined with TASC, full TFT achieves 57.47 on 3DREC Acc@0.25 and 45.37 mIoU, outperforming either component alone. We therefore believe the gains of TFT are not due to tuning alone, but are largely driven by the text-guided saliency in DGA, which strengthens fine-grained cross-modal alignment.
> # Weakness2 and Question2
> We thank the reviewer for recognizing the interest of DSG contribution and for helpful suggestion. We agree that comparisons with a broader range of learnable geometric alternatives would be valuable. However, spatial reasoning in current 3DVG is still mainly built on handcrafted priors such as Euclidean distances and angles, or on learnable weighting over predefined geometric bases rather than learning the relation basis itself. Given that this paradigm remains dominant, our contribution is to advance geometric modeling within this mainstream setting. We will clarify this scope in the final manuscript and discuss broader comparisons, such as GNN-based relational embeddings, as future work.
> # Weakness3 and Question3
> We thank the reviewer for this helpful suggestion and agree that the empirical positioning should be stated more carefully. In the final manuscript, we will revise the original statement “This indicates that DSGCR successfully…” to “This suggest that DSGCR is also effective in handling diverse spatial relations while maintaining competitive results across datasets.”
> # Weakness4 and Question4
> We thank the reviewer for raising this important concern regarding generalizability. To address it, we have conducted additional experiments by replacing the original 3DCLIP visual encoder with PointDif and Point-MaDi. As shown below, our proposed TFT and DSG consistently improve performance. PointDif + DSGCR brings gains of 1.46% on 3DREC Acc@0.25 and 0.81% mIoU, while Point-MaDi+DSGCR yields gains of 1.58% on 3DREC Acc@0.5 and 0.68% mIoU. Although PointDif and Point-MaDi adopt different pre-training paradigms, both benefit consistently from our designs. These results suggest that the improvements from TFT and DSG are not specific to 3DCLIP, but reflect gains in cross-modal alignment and geometric representation that transfer across pretrained visual encoders.
> | Backbone | 3DREC@0.25 | 3DREC@0.5 | 3DRES@0.25 | 3DRES@0.5 | mIoU |
> | :--- | :---: | :---: | :---: | :---: | :---: |
> | PointDif | 56.64 | 44.94 | 58.70 | 51.09 | 44.65 |
> | PointDif + DSGCR | **58.10** | **45.67** | **59.79** | **51.81** | **45.46** |
> | PointMadi | 56.95 | 44.78 | 59.01 | 51.16 | 44.88 |
> | PointMadi + DSGCR| **58.11** | **46.36** | **59.60** | **51.72** | **45.56** |
> # Weakness5 and Question5, Limitations
> We thank the reviewer for this helpful suggestion and agree that the limitations and failure cases should be discussed more explicitly. We acknowledge two main limitations. First, although DSGCR is effective on ScanRefer, Nr3D and Sr3D, its robustness on substantial larger-scale scenes or more diverse open-world settings needs further validation. Second, since current 3DVG methods mainly rely on handcrafted spatial priors, our geometric comparisons are also built on these standard baselines.Broader comparisons with other learnable geometric modeling methods, such as GNN-based ones, would be valuable future work.
>
> To provide clearer assessments, we will include **failure cases** in the final manuscript (Now shown in https://anonymous.4open.science/r/EE2DXX2002/Figure/failture_cases.png). (a)-(b) show viewpoint-sensitive failures, where model struggles to select correct viewpoint and grounds target to an incorrect object under ambiguous spatial descriptions. (c) shows inconsistency between the 3DREC box and 3DRES mask, suggesting that the two task heads do not always share a aligned intermediate representation  and that stronger cross-task consistency constraints may help. (d) shows that DSGCR tends to identify the primary spatial cue while neglecting secondary relational constraints. This indicates a limitation in our current spectral geometry modeling, which captures pairwise relations independently rather than reasoning over relational chains. (e) shows inaccurate segmentation boundaries and unreliable bounding box predictions when objects are occluded or represented by only a sparse set of points.
>
> Best regards,
>
> Wishing you a wonderful day!
>
> Authors

---

> > ### Author Rebuttal · Reviewer_oawD · 2026-04-04
> >
> > Thanks for the rebuttal. Most of my concerns have been properly addressed. I choose to maintain my positive score.

---

> > > ### Author Response · Authors · 2026-04-05
> > >
> > > Dear Reviewer oawD,
> > >
> > > Thank you for your careful review and positive assessment. We sincerely appreciate your acknowledgement that our rebuttal has properly addressed most of your concerns.
> > >
> > > Best regards,
> > >
> > > Authors

---

### Official Review · Reviewer_ACiz · 2026-03-13

**Soundness:** 2
**Presentation:** 1
**Significance:** 2
**Originality:** 3
**Overall Recommendation:** 3
**Confidence:** 4

**Summary:**

This paper addresses 3D visual grounding, including 3D referring expression comprehension (3DREC) and 3D referring expression segmentation (3DRES), which aim to localize or segment objects in 3D scenes based on natural language descriptions. The authors argue that existing methods often suffer from weak cross-modal alignment and limited geometric reasoning when modeling spatial relations in point clouds.

To address these issues, the paper proposes DSGCR, a framework designed to improve cross-modal semantic representation. The method introduces Text-aware Feature Tuning (TFT) to inject linguistic information into the visual encoder for better cross-modal alignment, and Decomposed Spectral Geometry (DSG) to model pairwise spatial relations using learnable Fourier features that decompose relations into symmetric and antisymmetric components.

The proposed approach is evaluated on ScanRefer, Nr3D, and Sr3D, covering both grounding and segmentation tasks. The reported results show improvements over prior methods on several metrics, and the paper includes ablation studies to analyze the contributions of the proposed components.

**Compliance With Llm Reviewing Policy:**

Affirmed.

**Final Justification:**

The authors have addressed most of my concerns and clarified several points. However, I remain concerned about the overall paper quality due to multiple evident citation errors and inaccuracies in the original submission, which affect the credibility of the work. While I am not able to give a positive recommendation, I will increase my score to 3, considering the authors’ responses.

**Key Questions For Authors:**

1. Motivation related to ShapeNet: The paper claims that features pretrained on ShapeNet are insensitive to semantic information and refers to Fig. 1(a), which shows a scene-level point cloud. Since ShapeNet is an object-level CAD dataset, could the authors clarify the connection between ShapeNet pretraining and the scene-level task focused in this paper?
2. Recent works increasingly leverage larger-scale pretrained models and LLM-based approaches for 3D grounding and multimodal reasoning. How do the authors view the trade-off between scaling pretraining/data and introducing handcrafted inductive biases such as the proposed designs? It would be helpful if the authors could discuss whether the proposed approach remains competitive when compared with methods that rely on stronger pretrained models or larger-scale data.

**Limitations:**

The paper includes an impact statement but provides only a very brief discussion and does not explicitly state the limitations of the proposed method. It would be helpful if the authors could include a clearer discussion of potential limitations or potential failure cases.

**Strengths And Weaknesses:**

**Strengths**
1. The paper proposes a structured framework that explicitly improves both cross-modal semantic alignment and geometric relation modeling, which are central challenges in 3D visual grounding.
2. The paper evaluates the proposed approach on multiple standard benchmarks, including ScanRefer, Nr3D, and Sr3D, covering both referring expression comprehension and segmentation tasks. The reported results also demonstrate consistent improvements over prior methods on several metrics across these datasets.
3. The paper includes well-designed and thorough ablation studies analyzing the contribution of the proposed modules, which helps provide insight into the role of each component.

**Weakness**
1. The sentence “Compared to 2D grounding…” in the introduction contains an incorrect citation. It cites SeeGround as an example of 2D grounding work, whereas SeeGround is a 3D visual grounding method.
2. The introduction claims that “As shown in Figure 1(a), visual features derived from generic 3D priors pretrained on ShapeNet (Li et al., 2021a) are often insensitive to semantic information”. However, this statement is problematic for two reasons. First, ShapeNet is a CAD object dataset, whereas Figure 1 illustrates an indoor scene point cloud, and the task studied in this paper is scene-level 3D visual grounding. As such, the example in Figure 1 does not meaningfully support the claim about ShapeNet pretraining, and the connection between ShapeNet and the problem setting of this paper is unclear. Second, the citation associated with “ShapeNet (Li et al., 2021a)” is incorrect and refers to an unrelated time-series classification paper, rather than the widely used 3D ShapeNet dataset. This mismatch weakens the credibility of the paper and suggests that the background discussion and references may not have been carefully verified.
3. The citation for RoBERTa (Zhuang et al., 2021) is incorrect. The referenced paper corresponds to PPBERT.
4. The experimental comparison on ScanRefer omits several prior methods that achieve higher scores on the “Unique” sub-metric such as M3DRef-CLIP, D-LISA, and ConcreteNet.
5. The ablation results in Tables 4–6 show only marginal performance changes across different variants. The improvements introduced by the proposed components are relatively small and inconsistent, making it difficult to clearly assess the actual contribution of each module.
6. The proposed approach relies on several handcrafted or inductive design choices. It is unclear whether such manually designed representations would generalize well to larger-scale and more diverse datasets, where learned representations may be more flexible and scalable. The paper does not provide evidence or discussion regarding the robustness of the proposed design under broader data distributions.

---

> ### Author Rebuttal · Authors · 2026-03-31
>
> Reviewer ACiz,
>
> # Weakness 1 and 3
> We sincerely apologize for the inadvertent citation errors. We will correct them in final manuscript: ① Remove the SeeGround citation from "Compared to 2D grounding..." and move it in the first sentence of the Introduction "3D Visual Grounding (3DVG)…". ② Correct the RoBERTa reference to “A robustly optimized BERT pretraining approach".
> # Weakness 2 and Question 1
> We thank the reviewer for this important observation. We agree that the original statement was imprecise and apologize for the confusion. Our visual encoder is pre-trained 3DCLIP (Hegde et al.,2023), as described in Sec.3.2. 3DCLIP is a 3D visual-language model pretrained by contrastive learning on point clouds, rendered images and texts. Although trained on ShapeNet, the resulting model has multimodal 3D priors rather than a purely object-level model. The original 3DCLIP paper also shows transferability to scene querying with language. We will revise the original statement and correct the citation as:"As shown in Figure 1(a), visual features derived from generic 3D visual-language priors (Hegde et al.,2023) are still insufficiently sensitive to fine-grained semantics.” We hope this revision conveys our motivation and clarifies the connection between ShapeNet pretraining and 3DVG.
> # Weakness 4
> We thank the reviewers for raising these comparisons. We respectfully argue that these methods differ fundamentally from our comparison scope about in paradigm and modality. Specifically, ConcreteNet is a dense 3D instance segmentation (3DRES) method that evaluates 3DREC by fitting boxes over predicted masks. This indirect evaluation differs from direct box regression methods in our table. For M3DRef-CLIP and D-LISA, they introduce online 2D multi-view rendering and employ 2D CLIP visual encoders to extract rich and appearance-based 2D features, which are advantageous for the Unique subset. Despite not leveraging 2D CLIP image features or mask-based box fitting, DSGCR still achieves the highest Overall Acc@0.5 of 46.97%, surpassing all three methods. Furthermore, our method demonstrates stronger disambiguation capability on the more challenging Multiple subset.
> # Weakness 5
> We respectfully note that in the highly challenging 3DVG, these absolute improvements are meaningful and not marginal. Compared with baseline, DSGCR achieves gains of 2.19%/2.43% in Acc@0.25/0.5. Given the 9508 validation targets, a 2.43% corresponds to more than 200 additional correctly localized objects under the stricter IoU, which is non-trivial in cluttered 3D scenes. Furthermore, the contributions of our modules are effective individually and complementary rather than inconsistent. TFT independently yields 1.18%/1.20% gain by improving fine-grained cross-modal alignment, while DSG independently contributes 1.62%/0.95% by enhancing geometric reasoning and reducing spatial ambiguity.
>
> Regarding Tables 5 and 6, the purpose of ablations is to verify the role of each sub-component within each module. In Table 5, both TASC and DGA outperform the baseline and their combination gives the best result, demonstrating that stable feature calibration and explicit text-guided gating are both important for effective cross-modal alignment. In Table 6, both the symmetric and antisymmetric terms are helpful and full DSG performs best, confirming that proximity-aware and direction-aware relations are necessary for complex 3DVG.
>
> # Weakness 6 and Question2
> We thank the reviewer for these insightful comments regarding generalizability and positioning relative to larger-scale pretrained models. We will incorporate this discussion into the limitations section of final manuscript.
>
> We would like to respectfully note DSG is not a purely manually method. The Fourier frequency matrix and nonlinear MLP are learnable. Meanwhile, DSG does include inductive bias, since the symmetric/antisymmetric decomposition is motivated by directional and relational language common in 3DVG, such as left/right. While effective across ScanRefer, Nr3D and Sr3D, we agree that its robustness under larger-scale or more diverse data remains to be fully validated.
>
> We view larger-scale pretraining/LLM-based methods and inductive biases as complementary. In practice, many LLM/MLLM-based 3D grounding methods rely on task-specific prompting, decomposition or reasoning pipelines, which suggests that inductive bias remains important even in scaling-based methods. While stronger pretrained models provide richer general semantic priors, 3DVG still requires precise spatial reasoning, especially for directional and non-commutative relations. Our DSG helps alleviate this gap by a spectral decomposition of distance-aware and directional relations. Moreover, DSG is designed as a lightweight, architecture-agnostic plug-in module and could potentially enhance spatial attention in other models.
> # Failure cases
> Please see response to Reviewer oawD Weakness5
>
>
> Best regards,
>
> Wishing you a wonderful day!
>
> Authors

---

> > ### Author Rebuttal · Reviewer_ACiz · 2026-04-02
> >
> > The authors have addressed most of my concerns and clarified several points. However, I remain concerned about the overall paper quality due to multiple evident citation errors and inaccuracies in the original submission, which affect the credibility of the work. While I am not able to give a positive recommendation, I will increase my score to 3, considering the authors’ responses.

---

> > > ### Author Response · Authors · 2026-04-03
> > >
> > > Dear Reviewer ACiz,
> > >
> > > We sincerely thank you for raising your score after acknowledging that our rebuttal had addressed most of your concerns.
> > >
> > > We fully understand your concern regarding the citation and imprecise expressions in the original submission, and we sincerely apologize for these inadvertent issues. **As stated in our rebuttal, we have committed to incorporating the following corrections into the revised manuscript:**
> > >
> > > # weakness1,3 - Citations
> > >  **① Remove the SeeGround citation from the sentence "Compared to 2D grounding..." and move it appropriately in the first sentence of the Introduction "3D Visual Grounding (3DVG)…".**
> > >
> > >  **② Correct the RoBERTa reference to “A robustly optimized BERT pretraining approach".**
> > >
> > > # weakness2 - Expression
> > >
> > > Based on our clarification of the relationship between ShapeNet pretraining and scene-level 3DVG，We will revise the original statement and update expression into  the revised manuscript: **"As shown in Figure 1(a), visual features derived from generic 3D visual-language priors (Hegde et al.,2023) are still insufficiently sensitive to fine-grained semantics.”**
> > >
> > > Beyond the specific corrections already committed to in our rebuttal to Weaknesses 1,2 and 3, we have also conducted a line-by-line manual audit of all citations and references to eliminate any remaining inconsistencies.  We will ensure that the final version meets the expected standards of rigor in both presentation and substance.
> > >
> > > Best regards,
> > >
> > > Authors

---

### Decision · Program_Chairs · 2026-04-30

**Decision:**

Accept (regular)

**Comment:**

The reviewers initially had concerns regarding limited validation of generalizability as well as citation errors and presentation inaccuracies. The authors' rebuttals successfully resolved most of the reviewers' concerns. However, one of the reviewers still remained concerned about the overall paper quality due to multiple citation errors and inaccuracies in the original submission. The AC read all reviews and rebuttals, and appreciate the novel mathematical framework and strong experimental results. Finally, the AC reached the conclusion that the overall contributions of this paper warrant its acceptance.